

# Chiral magnetism: a geometric perspective

**Daniel Hill[1], Valeriy Slastikov[2] and Oleg Tchernyshyov[1⋆]**

**1** Department of Physics and Astronomy and Institute for Quantum Matter,
Johns Hopkins University, Baltimore, MD 21218, USA
**2** School of Mathematics, University of Bristol, Bristol BS8 1TW, UK

⋆ olegt@jhu.edu

## Abstract

We discuss a geometric perspective on chiral ferromagnetism. Much like gravity becomes the effect of spacetime curvature in theory of relativity, the Dzyaloshinski-Moriya interaction arises in a Heisenberg model with nontrivial spin parallel transport. The Dzyaloshinski-Moriya vectors serve as a background SO(3) gauge field. In 2 spatial dimensions, the model is partly solvable when an applied magnetic field matches the gauge curvature. At this special point, solutions to the Bogomolny equation are exact excited states of the model. We construct a variational ground state in the form of a skyrmion crystal and confirm its viability by Monte Carlo simulations. The geometric perspective offers insights into important problems in magnetism, e.g., conservation of spin current in the presence of chiral interactions.

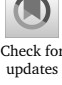

# 1 Introduction

## 1.1 The specific problem: the skyrmion crystal

Chiral magnetic order, exemplified by helicoidal and more complex periodic structures, has a long history in the field of magnetism. These periodic spatial modulations arise from a competition of the Heisenberg exchange and of a weaker Dzyaloshinskii–Moriya (DM) interaction induced by the relativistic spin-orbit coupling [1,2]. In recent years, chiral magnetism has received renewed interest in connection with the experimental discovery [3] of the skyrmion crystal, a magnetic analog of the Abrikosov vortex lattice [4] predicted by Bogdanov and Yablonskii [5]. Both superconducting vortices and magnetic skyrmions are examples of topological solitons. Even the simplest theories allowing for such soliton lattices have a strongly nonlinear character, which makes finding analytical solutions a highly nontrivial problem [6]. Whereas Abrikosov found an exact solution for a vortex lattice in extreme type-II superconductors, no such feat has been accomplished for a skyrmion crystal in models of chiral magnetism to the best of our knowledge.

In this paper we discuss a simple model of chiral magnetism that comes close to this goal. We build on the classic work of Belavin and Polyakov [7], who found an entire class of exact excited states for the pure Heisenberg model in $d = 2$ dimensions. Such special states occur in many nonlinear field theories and are known today as Bogomolny solutions [8]. Their stability is topological in nature: a Bogomolny state minimizes the energy in its topological sector. Furthermore, the energy of such a state is proportional to the corresponding topological charge $Q$. In the Heisenberg model, the Bogomolny lower bound for the energy is

$$E = 4\pi Q, \tag{1}$$

where $Q$ is the skyrmion number.

Table 1: Analogy between theories of gravity and chiral magnetism.

| General relativity | Chiral magnetism |
|---|---|
| particle's 4-velocity $u^i$ | magnetization $m_\alpha$ |
| acceleration $du^i/d\tau$ | magnetization twist $\partial_i m_\alpha$ |
| Levi-Civita connection $\Gamma^i{}_{jk}$ | spin connection $a_{i\alpha\beta}$ |
| Riemann curvature $R^i{}_{jkl}$ | spin curvature $f_{ij\alpha\beta}$ |

This approach cannot be used directly to obtain the exact ground state of the gauged Heisenberg model because the ground state lives in a topological sector without Bogomolny states. Nonetheless, a study of Bogomolny states in other sectors provides us with enough clues to construct a variational ground state. The idea is to examine states with a negative skyrmion number $Q$ made from skyrmion antiparticles, which we call antiskyrmions.[1]

The lack of Bogomolny solutions in topological sectors with $Q < -1$ tells us that antiskyrmions behave as interacting particles because their energy is not given by the Bogomolny lower bound (1). An examination of variational states with two antiskyrmions reveals repulsive interactions slowly decaying with the distance. The construction of a variational ground state then begins with a collection of well-separated antiskyrmions. In this limit, the interactions may be neglected; then each antiskyrmion contributes $-4\pi$ to the energy. As we add more antiskyrmions, the energy is at first lowered in proportion to the number of added antiparticles, $E \approx 4\pi Q$. However, as antiskyrmions become more dense, their repulsive interactions strengthen, making the addition of each new antiparticle less advantageous. At some optimal concentration of antiskyrmions, the energy reaches a minimum and we get our ground state. Our calculations show that the minimum is reached when the antiskyrmions are still far apart and their interactions are relatively weak, so that our picture of antiskyrmions as weakly interacting particles is applicable.

## 1.2 The broader impact: geometrization of chiral magnetism

Although our immediate goal is to solve a specific problem, we would like to call the attention of the reader to the theoretical framework we used. In our view, the general method employed in this paper is more interesting than the specific narrow task to which it has been applied. This perspective is not exactly new—it can be traced to a 1978 work of Dzyaloshinskii and Volovik [10]. It has recently received renewed attention from Schroers and collaborators [11–13] but remains largely unknown among condensed-matter physicists. The beauty and promise of this theoretical approach compels us to lay it out in some detail in this paper.

The general framework used here can be characterized as the geometrization of chiral interactions. Whereas in the commonly accepted view chiral magnetic order is a result of competing physical *interactions*, in the alternative picture it is an outcome of a modified *geometry* of spin parallel transport. This approach is entirely similar in spirit to the geometrization of gravity in Einstein's theory of relativity [14]. Much like general relativity describes the motion of a particle in an arbitrary, possibly accelerating, reference frame, geometric theory of chiral magnetism quantifies spin in a local spin frame that twists in space. A spatial inhomogeneity of magnetization components $m_\alpha(x)$ is akin to a variation of a particle's 4-velocity $u^i(\tau)$ in proper time.[2] The analogy is summarized in Table 1.

---

[1] We use the term "antiskyrmion" as a shorthand for solitons with skyrmion number $Q = -1$. This is different from the convention introduced by Koshibae and Nagaosa [9].

[2] Throughout the paper, Greek indices label spin components; Latin indices label spatial coordinates.

The geometric approach offers a surprising perspective on the DM interaction. The DM vectors, quantifying chiral energy in the traditional approach, turn out to be the spin connection, or the SO(3) gauge potential, i.e., a frame-dependent, and therefore unphysical, auxiliary quantity. It can even be gauged away for any single spatial direction by a judicious choice of the local spin frame.

The formal basis for the geometric perspective is an extension of the Heisenberg model, whose energy (13a) is invariant under *global* rotations of the spin reference frame, to a gauged version of the same model, whose energy is invariant under *local* spin-frame rotations. The analog in relativity theory is the extension from special relativity with Lorentz transformations to general relativity with arbitrary coordinate transformations.

The rest of the paper is organized as follows. We introduce the reader to the gauged version of the Heisenberg model in Sec. 2, where we emphasize a geometric perspective rooted in the notion of spin parallel transport and illustrate its utility by extending the concept of conserved spin current to situations with chiral interactions that break the global spin-rotation symmetry. In Sec. 3 we present the main result of our work: the determination of the ground state of the gauged Heisenberg model in $d = 2$ dimensions at its solvable point. We construct a variational ground state and confirm its viability by Monte-Carlo simulations. To keep the narrative focused, technical discussions are collected in the appendixes.

## 2 Chiral magnetism: a geometric perspective

In this section we introduce a covariant version of the Heisenberg model that allows for arbitrary rotations of the local spin frame. A difference in the orientations of spin frames at adjacent points requires the introduction of a spin connection, or the SO(3) gauge field. Although an SO($N$) gauge field is usually matrix-valued, an equivalent vector formulation is possible for $N = 3$. We choose the vector formulation because this language is used in chiral magnetism. A translation between the matrix and vector languages is given in Appendix A.1.

### 2.1 Spin vectors

A vector of spin is specified by its three components $(S_1, S_2, S_3)$ in a Cartesian spin frame. A rotation of the frame through an infinitesimal angle $(\omega_1, \omega_2, \omega_3)$ changes the spin components as follows:

$$\delta S_\alpha = -\epsilon_{\alpha\beta\gamma}\omega_\beta S_\gamma, \tag{2}$$

where $\epsilon_{\alpha\beta\gamma}$ is the antisymmetric Levi-Civita symbol in the spin space. To streamline the notation, we will use a shorthand $\mathbf{S} \equiv (S_1, S_2, S_3)$ and write the law of component transformation (2) in a more compact vector form:

$$\delta\mathbf{S} = -\boldsymbol{\omega} \times \mathbf{S}. \tag{3}$$

The reader should keep in mind that the rotation of a spin frame is a *passive* transformation. It does not alter the physical state of a spin but merely changes its description in terms of Cartesian components.

Any quantity whose components transform under spin-frame rotations according to Eqs. (2) and (3) will be referred to a *spin vector*. Spin vectors will be set in the boldface type. Examples of spin vectors in this paper are magnetization $\mathbf{m}$, rotation angle $\boldsymbol{\omega}$, and spin current flowing in along the $x_i$-axis $\mathbf{j}_i$.

Note that spin-frame rotations do not affect spatial coordinates. Therefore the shorthand for spatial coordinates $x \equiv (x_1, \ldots, x_d)$ is not in the boldface type.

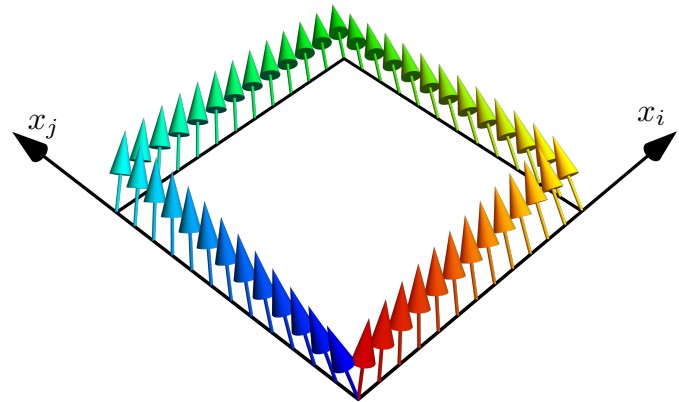

Figure 1: A spin is transported along a rectangular loop in the $(x_i, x_j)$ plane with sides $dx_i$ and $dx_j$. The initial (red) and final (blue) spin orientations differ by a rotation through the angle $\mathbf{F}_{ij}\, dx_i \wedge dx_j$.

## 2.2 Local rotations and the SO(3) gauge field

A complication arises when we consider a vector field such as magnetization $\mathbf{m}(x)$ and wish to compare its values at two different points. If the spin frames at these points have different orientations then it does not make sense to compare their components directly.

This problem can also be seen from the perspective of rotational symmetry. Although the magnetization field $\mathbf{m}(x)$ transforms under a local infinitesimal rotation in the proper way (3),

$$\delta\mathbf{m}(x) = -\boldsymbol{\omega}(x) \times \mathbf{m}(x), \tag{4}$$

its spatial derivative $\partial_i \mathbf{m}(x)$ does not,

$$\delta\partial_i\mathbf{m}(x) = -\boldsymbol{\omega}(x) \times \partial_i\mathbf{m}(x) - \partial_i\boldsymbol{\omega}(x) \times \mathbf{m}(x) \neq -\boldsymbol{\omega}(x) \times \partial_i\mathbf{m}(x),$$

if the rotation angle $\boldsymbol{\omega}(x)$ varies in space. Therefore $\partial_i\mathbf{m}(x)$ is not a spin vector.

The symmetry perspective is particularly useful because it can offer a solution: fix the notion of the derivative so that it would behave as a spin vector. That is the covariant derivative

$$D_i\mathbf{m}(x) \equiv \partial_i\mathbf{m}(x) - \mathbf{A}_i(x) \times \mathbf{m}(x). \tag{5}$$

Here $\mathbf{A}_i(x)$ is the spin connection, or the SO(3) gauge field, for direction $x_i$. The covariant derivative transforms as a vector (3), provided that the gauge field transforms as follows:

$$\delta\mathbf{A}_i(x) = -D_i\boldsymbol{\omega}(x). \tag{6}$$

Note that the gauge field $\mathbf{A}_i(x)$ is not a spin vector: the gauge transformation (6) generally differs from the transformation of vectors (3).

## 2.3 Spin parallel transport and curvature

The condition

$$D_i\mathbf{m}(x) = 0 \tag{7}$$

defines the rule of parallel transport for spin vector $\mathbf{m}$. Starting with vector $\mathbf{m}(x)$ at point $x$, we end with vector

$$\mathbf{m}(x + dx) = \mathbf{m}(x) + \mathbf{A}_i(x) \times \mathbf{m}(x)\, dx_i \tag{8}$$

at a neighboring point $x + dx$.

Here is a trivial example where a gauge field is required for the description of parallel transport. Imagine that a spin vector is moving through space without changing its orientation relative to a *global* spin frame. However, the *local* spin frame varies from point to point.[3] To compensate for the spatial twists of the local spin frame, spin components $m_\alpha$ must vary in space as well and have nonzero spatial gradients: $\partial_i m_\alpha = \epsilon_{\alpha\beta\gamma} A_{i\beta} m_\gamma$, where the gauge field $\mathbf{A}_i$ is determined by the spatial twists of the local spin frame. See Appendix A.1.

A nontrivial example would be a spin whose physical orientation changes relative to a global spin frame as it moves through space. Even if the local spin frame has a fixed orientation and does not change from point to point, the spin coordinates $m_\alpha$ will vary in space, reflecting an actual, physical twist of the transported spin. See Appendix A.2.

In the two examples above, we had to assume the existence of an absolute (global) spin frame, against which the twists of the local spin frame or of the spin itself can be measured. However, just like in general relativity, the existence of an absolute reference frame is unnecessary. We can make do with local spin frames entirely.

To determine whether or not parallel transport is trivial without relying on a global spin frame, we take a spin around a closed loop in space and check whether the spin retains its original orientation at the end of the trip. In Fig. 1, the loop is an infinitesimal rectangle in the plane $(x_i, x_j)$ with sides $dx_i$ and $dx_j$ and an oriented area $dx_i \wedge dx_j = -dx_j \wedge dx_i$. A spin transported around the loop is rotated by the angle

$$\mathbf{\Omega} = \mathbf{F}_{ij}\, dx_i \wedge dx_j \quad \text{(no sum over } i \text{ or } j \text{ here),} \tag{9}$$

where

$$\mathbf{F}_{ij} = \partial_i \mathbf{A}_j - \partial_j \mathbf{A}_i - \mathbf{A}_i \times \mathbf{A}_j \tag{10}$$

is the curvature of the gauge field. $\mathbf{F}_{ij}$ is a spin vector: it transforms under rotations of the spin frame in the standard way (3). Parallel transport is nontrivial if $\mathbf{F}_{ij} \neq 0$.

The covariant derivative possesses most properties of the ordinary derivative. For example, integration by parts can be done in the familiar way, with the aid of the identity

$$\mathbf{m} \cdot D_i \mathbf{n} = -\mathbf{n} \cdot D_i \mathbf{m} + D_i (\mathbf{m} \cdot \mathbf{n}). \tag{11}$$

The covariant derivative of a scalar is simply the ordinary derivative, $D_i f \equiv \partial_i f$.

One important distinction of the covariant derivative is its noncommutative nature:

$$D_i D_j \mathbf{m} - D_j D_i \mathbf{m} = -\mathbf{F}_{ij} \times \mathbf{m}. \tag{12}$$

## 2.4 Gauged Heisenberg model

The standard model of chiral magnetism is a continuum theory where the local spin or magnetization is represented by a 3-component vector field $\mathbf{m}(x)$ of unit length, the competing Heisenberg exchange and DM energies are

$$U_{\text{ex}} = \frac{1}{2} \int d^d r\, \partial_i \mathbf{m} \cdot \partial_i \mathbf{m}, \tag{13a}$$

$$U_{\text{DM}} = -\int d^d r\, \mathbf{d}_i \cdot (\mathbf{m} \times \partial_i \mathbf{m}). \tag{13b}$$

The units of energy and length are chosen so that the exchange interaction (13a) has unit strength. The DM vectors $\mathbf{d}_i$ in Eq. (13b) have the dimension of inverse length and determine the wavenumber of incommensurate magnetic order.

---

[3]The analog in general relativity would be the uniform motion of a body described in a non-inertial reference frame. Spatial inhomogeneity of spin components, $\partial_i m_\alpha \neq 0$, translates to a time variation of velocity, $dv^i/dt \neq 0$.

Replacing the ordinary spatial derivatives in the Heisenberg exchange energy (13a) with the covariant ones yields the gauged Heisenberg model,

$$U_{\text{gauged}} = \frac{1}{2} \int d^d r \, D_i \mathbf{m} \cdot D_i \mathbf{m}. \tag{14}$$

Whereas the regular Heisenberg model (13a) favors a uniform magnetization field, $\partial_i \mathbf{m}(x) = 0$, its gauged version (14) prefers an $\mathbf{m}(x)$ that follows the rules of parallel transport (7). A field configuration with zero energy can be constructed by starting with some $\mathbf{m}(0)$ at the origin and using parallel transport to obtain $\mathbf{m}(x)$ at every other point $x$. Different paths leading from 0 to $x$ must yield the same $\mathbf{m}(x)$, so this procedure is only possible if the parallel transport is trivial, $\mathbf{F}_{ij} = 0$. For $\mathbf{F}_{ij} \neq 0$, no configuration of the field can have $D_i \mathbf{m} = 0$ everywhere, for it would imply that $D_i D_j \mathbf{m} - D_j D_i \mathbf{m} = D_i 0 - D_j 0 = 0$ in contradiction with Eq. (12). The energy (14) is therefore strictly positive even in a ground state. Finding ground states of a gauged Heisenberg model with $\mathbf{F}_{ij} \neq 0$ is therefore a nontrivial problem.

The gauged Heisenberg model (14) has a natural connection to chiral ferromagnetism. Let us examine its energy by orders in the gauge field $\mathbf{A}_i$. To the zeroth order, the gauged theory (14) reduces to the pure Heisenberg model (13a). The first-order term $-\int d^d r \, \partial_i \mathbf{m} \cdot (\mathbf{A}_i \times \mathbf{m})$ yields the DM energy (13b), provided that we set $\mathbf{A}_i = \mathbf{d}_i$. At this order of the expansion, the gauged Heisenberg model reproduces the basic chiral model (13). It is remarkable that the Dzyaloshinskii-Moriya interaction is now encoded in the spin geometry, wherein the DM vectors serve as the SO(3) gauge field.

At the second, and final, order in $\mathbf{A}_i$, the gauged theory has an anisotropy term,

$$U_{\text{an}} = \frac{1}{2} \int d^d r \, (\mathbf{d}_i \times \mathbf{m}) \cdot (\mathbf{d}_i \times \mathbf{m}) = -\frac{1}{2} \int d^d r \, (\mathbf{d}_i \cdot \mathbf{m})(\mathbf{d}_i \cdot \mathbf{m}) + \text{const.} \tag{15}$$

The addition of this term is not necessarily a problem. In a magnet with a cubic symmetry such as MnSi [15] this correction yields an $\mathbf{m}$-independent constant and the gauged model (14) is equivalent to the chiral one (13). In magnets with a lower symmetry the anisotropy term (15) does not generally cancel out. However, its presence can be justified. When the Heisenberg exchange (13a) and DM interaction (13b) are derived from a simple microscopic model—such as Hubbard's—exactly the right anisotropy term (15) arises as well [16]. This is not a coincidence but a reflection of a deeper principle. The spin-orbit coupling, too, can be expressed as an SU(2) gauge field acting on the electron spinor wavefunction [17, 18]; that gauge field turns into the SO(3) gauge field in the effective spin model. There are, of course, additional sources of spin anisotropy—such as long-range dipolar interactions—that are not captured by the gauged Heisenberg model (14).

## 2.5 Spin conservation

To appreciate the geometric perspective on chiral magnetism, we turn to a conservation law associated with the symmetry of spin rotations.

### 2.5.1 Pure Heisenberg model

The energy of the pure Heisenberg model (13a) is invariant under global rotations, $\mathbf{m}(x) \mapsto R^{-1} \mathbf{m}(x)$, where an SO(3) matrix $R$ represents the rotation of the spin frame.[4] This symmetry ensures the conservation of total spin $\mathbf{S} = \mathcal{S} \int d^d x \, \mathbf{m}$, where $\mathcal{S}$ is the spin length per unit

---

[4]Recall that these rotations are passive transformations. The spin frame and components of the spin transform, but the physical state of the spin remains unchanged.

volume. This global symmetry gives rise to a local conservation law, the continuity of spin current:

$$\partial_t \mathbf{s} + \partial_i \mathbf{j}_i = 0. \tag{16}$$

Here $\mathbf{s}$ is the spin density and $\mathbf{j}_i$ is the spin current flowing along the $x_i$-axis.

Although the continuity equation (16) follows directly from Noether's theorem, its standard field-theoretic derivation is complicated by a gauge dependence of the Berry-phase term in the action of a ferromagnet [19,20]. Instead, we can be obtain it from the Landau–Lifshitz equation of motion for the magnetization field,

$$\mathcal{S}\partial_t \mathbf{m} = -\mathbf{m} \times \frac{\delta U}{\delta \mathbf{m}}. \tag{17}$$

The left-hand side of Eq. (17) is the time derivative of the spin density $\mathbf{s} = \mathcal{S}\mathbf{m}$. The right-hand side is the conservative torque density derived from an energy functional $U$. If the latter contains exchange energy (13a) and nothing else then the torque density can be written as a divergence: $-\mathbf{m} \times \delta U_{\mathrm{ex}}/\delta \mathbf{m} = \mathbf{m} \times \partial_i \partial_i \mathbf{m} = \partial_i(\mathbf{m} \times \partial_i \mathbf{m})$. Comparison with the continuity equation (16) immediately yields the expression for spin current flowing along the $x_i$-axis,

$$\mathbf{j}_i = -\mathbf{m} \times \partial_i \mathbf{m}. \tag{18}$$

Anisotropic interactions violate the global SO(3) symmetry and spoil spin conservation. In the presence of DM interactions (13b), the spin current is no longer conserved:

$$\partial_t \mathbf{s} + \partial_i \mathbf{j}_i = -\mathbf{m} \times \frac{\delta U_{\mathrm{DM}}}{\delta \mathbf{m}} = -2\mathbf{m} \times (\mathbf{d}_i \times \partial_i \mathbf{m}) \neq 0. \tag{19}$$

We thus find that the conservation of spin current (16) breaks down already at the first order in the relativistic spin-orbit coupling.

### 2.5.2 Gauged Heisenberg model

The gauged Heisenberg model (14) is invariant under a larger symmetry group of local spin rotations, $\mathbf{m}(x) \mapsto R^{-1}(x)\mathbf{m}(x)$. Following Chandra *et al.* [21], we exploit this local version of the spin-rotation symmetry to find the conserved spin current in the gauged Heisenberg model. As in Sec. 2.5.1, we derive it from the equation of motion for magnetization, which now reads

$$\partial_t \mathbf{s} = -\mathbf{m} \times \delta U_{\mathrm{gauged}}/\delta \mathbf{m} = \mathbf{m} \times D_i D_i \mathbf{m} = -D_i \mathbf{j}_i. \tag{20}$$

Here

$$\mathbf{j}_i = -\mathbf{m} \times D_i \mathbf{m} \tag{21}$$

is the redefined, gauge-invariant spin current. The local law of spin-current conservation now includes the covariant spatial derivatives,

$$\partial_t \mathbf{s} + D_i \mathbf{j}_i = 0. \tag{22}$$

In principle, the time derivative can also be gauged [21], but we have no need for that in the present work.

The advantage of the spin-current conservation in its gauged form (20) is that the dominant source of anisotropy—the DM interaction—is included in the definition of spin current. The spin current is automatically conserved to the first order in the relativistic spin-orbit coupling. Any additional anisotropy beyond the already included term (15) will be of the second order in the spin-orbit coupling, so any further violation of spin-current conservation will be comparatively minor.

## 2.6 Historical note

The first use of the covariant derivative (5) in magnetism dates back to a 1978 work of Dzyaloshinskii and Volovik [10] on the Heisenberg spin glass. Chandra *et al.* [21] used it for a theory of an antiferromagnet whose magnetic order has been destroyed by quantum fluctuations. They also extended the notion of conserved spin current to situations where the global SO(3) symmetry is replaced with a local version.

Despite its natural connection to chiral magnetism, the gauged Heisenberg model has not been widely used in this context. The 1992 work of Shekhtman *et al.* [16] deserves a special mention, even though it barely mentions the word "gauge." It is nonetheless a lattice version of the gauge theory and its primary result can be formulated most succinctly in the geometric language: an antiferromagnet with chiral interactions has no net magnetic moment if the SO(3) gauge field has zero curvature. Gaididei and collaborators [22–24] used the geometric approach to study the effects of spatial curvature in thin magnetic films.

Most recently, Schroers [11] described a chiral ferromagnet in terms of an SO(3) gauge theory. For a special value of an applied magnetic field, equal to the curvature of the SO(3) gauge field, the model has a large class of exact solutions that minimize the energy locally. Schroers and collaborators [12, 13] found these solutions by adopting the method of Belavin and Polyakov [7]. This class of solutions has skyrmions acting as noninteracting particles with the energy of $4\pi$ each. A diverse array of multi-skyrmion configurations was constructed analytically and confirmed numerically [25]. The absolute ground states of the model are, unfortunately, not among these special states. Ross *et al.* [26] extended these efforts beyond the exactly solvable point. They showed that two skyrmions interact via a repulsive potential decaying as a power of the distance. By tuning the anisotropy term, they found regimes where the energy of an individual skyrmion could be made negative. A viable candidate for a ground state would then be a dense skyrmion lattice, where the skyrmion repulsion balances the negative skyrmion energy.

The geometric approach has also been applied to the magnetism of conduction electrons. This perspective was introduced by Frölich and Studer [17], who showed that the spin-orbit coupling can be recast as parallel transport of the spinor wavefunction in a background SU(2) gauge field. Tokatly [18] pointed out that the gauge perspective offers a simple way to define a conserved spin current in the presence of spin-orbit coupling and computed its equilibrium value for the Rashba and Dresselhaus models.

For more recent applications of the gauge theory to magnetism see Refs. [27–31].

## 3   Skyrmion crystal in a two-dimensional chiral ferromagnet

To demonstrate the utility of the gauged Heisenberg model, we have studied it in 2 spatial dimensions. Our analytical arguments and numerical simulations strongly suggest that the ground state of the model is a hexagonal skyrmion crystal at least for a magic value of an applied magnetic field and possibly beyond. The skyrmion lattice is a notoriously fickle magnetic phase of matter that requires finely tuned temperature and applied magnetic field [3].

In this section we consider the gauged Heisenberg model with an applied magnetic field, whose energy functional is

$$U = \int_{\Omega} d^2x \left( \frac{1}{2} D_i \mathbf{m} \cdot D_i \mathbf{m} - \mathbf{h} \cdot \mathbf{m} \right). \tag{23}$$

The background gauge fields are equal to the Dzyaloshinskii-Moriya vectors, $\mathbf{A}_i = \mathbf{d}_i$, whose values are constrained by the symmetry of the magnetic material [5]. For symmetry classes

Table 2: Choice of the gauge fields $\mathbf{A}_i = \mathbf{d}_i$, the resulting gauge curvature $\mathbf{F}_{12}$, and the Bogomolny equation for the upper signs in Eqs. (29) and (31).

| Symmetry class | $\mathbf{A}_1$ | $\mathbf{A}_2$ | $\mathbf{F}_{12}$ | Bogomolny equation |
|:---:|:---:|:---:|:---:|:---:|
| $C_{nv}$ | $+\kappa \mathbf{e}^{(2)}$ | $-\kappa \mathbf{e}^{(1)}$ | $-\kappa^2 \mathbf{e}^{(3)}$ | $\partial \bar{\chi} / \partial \bar{z} = \kappa/2$ |
| $D_n$ | $+\kappa \mathbf{e}^{(1)}$ | $+\kappa \mathbf{e}^{(2)}$ | $-\kappa^2 \mathbf{e}^{(3)}$ | $\partial \bar{\chi} / \partial \bar{z} = i\kappa/2$ |
| $D_{2d}$ | $-\kappa \mathbf{e}^{(1)}$ | $+\kappa \mathbf{e}^{(2)}$ | $+\kappa^2 \mathbf{e}^{(3)}$ | $\partial \psi / \partial \bar{z} = i\kappa/2$ |

$C_{nv}$, $D_n$ ($n = 3, 4, 6$) and $D_{2d}$, the directions of the DM vectors are fixed and the only choice is the overall magnitude $\kappa$, Table 2. Here $\mathbf{e}^{(1)}$ and $\mathbf{e}^{(2)}$ are unit vectors in the plane of the film and $\mathbf{e}^{(3)} = \mathbf{e}^{(1)} \times \mathbf{e}^{(2)}$. We will use the symmetry class $D_n$ as a specific example in what follows, but the method is readily extendable to the other cases.

The model with the energy functional (23) is partially solvable for a magic value of the applied magnetic field set by the curvature of the spin parallel transport,

$$\mathbf{h} = \mp \mathbf{F}_{12}. \tag{24}$$

At this value of the applied field, the model has a large class of stationary solutions $\mathbf{m}(x)$ of the equation of motion (17) that minimize the energy locally. Because Eq. (17) is a nonlinear partial differential equation (PDE) of the second order, finding stationary states in nonlinear field theories is generally a hard problem. However, some field theories have special classes of field configurations, known as Bogomolny states, that satisfy a simpler first-order PDE and saturate a lower bound on the energy determined by a topological invariant [8, 32]. In the context of magnetism, the Bogomolny states were found by Belavin and Polyakov in the pure Heisenberg model in two dimensions [7]. The Bogomolny states for the gauged Heisenberg model (23) are directly related to them, so we briefly review their construction.

### 3.1 Bogomolny states in the pure Heisenberg model

Local energy minima of the pure Heisenberg model (13a) satisfy a second-order PDE

$$0 = \mathbf{m} \times \frac{\delta U_{\text{ex}}}{\delta \mathbf{m}} = -\mathbf{m} \times \partial_i \partial_i \mathbf{m}. \tag{25}$$

In view of the constraint $|\mathbf{m}(x)| = 1$, this PDE is nonlinear. Aside from the simple uniform ground states $\mathbf{m}(x) = \text{const}$, its solutions are hard to find.

Field configurations satisfying a simpler first-order PDE, the Bogomolny equation

$$\partial_1 \mathbf{m} \pm \mathbf{m} \times \partial_2 \mathbf{m} = 0, \tag{26}$$

saturate the lower bound for the energy,

$$E \geq E_{\text{min}} = \pm \int_{\Omega} d^2 x \, \mathbf{m} \times (\partial_1 \mathbf{m} \times \partial_2 \mathbf{m}) \equiv \pm 4\pi Q, \tag{27}$$

and are therefore local energy minima. Here the topological charge $Q$ is the skyrmion number. Thus skyrmions turn out to be elementary excitations of the Heisenberg model in $d = 2$ spatial dimensions with the energy $4\pi$. In what follows, we will stick with the upper signs in Eqs. (26) (27) and their gauged counterparts. The lower signs correspond to time-reversed situations.

The Bogomolny solutions of the pure Heisenberg model (13a) are most efficiently represented in terms of complex coordinates and fields,

$$
\begin{aligned}
z &= x^1 + ix^2, & \bar{z} &= x^1 - ix^2, \\
\psi &= \frac{m_1 + im_2}{1 + m_3}, & \bar{\psi} &= \frac{m_1 - im_2}{1 + m_3}, \\
\chi &= \frac{-m_1 - im_2}{1 - m_3}, & \bar{\chi} &= \frac{-m_1 + im_2}{1 - m_3}.
\end{aligned}
\tag{28}
$$

Here $\chi \equiv -1/\bar{\psi}$ and $\bar{\chi} \equiv -1/\psi$ are time-reversed copies of $\psi$ and $\bar{\psi}$ introduced for notational convenience. The Bogomolny equation for the upper sign in Eqs. (26) reads $\partial \psi / \partial \bar{z} = 0$, so its solutions $\psi = w(z)$ are arbitrary meromorphic functions of $z$ [7]. (The same applies to $\bar{\chi}$.) The skyrmion number is the degree of mapping $z \mapsto w(z)$, taking on values $Q = 0, 1, 2, \ldots$ Uniform states, $\psi = \text{const}$, yield the lowest possible $Q = 0$ and $E = 0$. States $\psi = \prod_{n=1}^{N}(z - z_n)$ and $\psi = \sum_{n=1}^{N} 1/(z - z_n)$ have $N$ skyrmions at complex positions $z_n$ and $E = 4\pi N$. Thus skyrmions behave like noninteracting particles with energy $4\pi$.

## 3.2 Bogomolny states in the gauged Heisenberg model

The Bogomolny equation of the gauged Heisenberg model (23) is

$$
D_1 \mathbf{m} \pm \mathbf{m} \times D_2 \mathbf{m} = 0.
\tag{29}
$$

Bogomolny states saturate the lower energy bound

$$
E \geq E_{\min} = \int_{\Omega} d^2 x \left[ -\mathbf{h} \cdot \mathbf{m} \pm \mathbf{m} \cdot (D_1 \mathbf{m} \times D_2 \mathbf{m}) \right].
\tag{30}
$$

At the solvable point (24), the energy bound has the following form [11]:

$$
E_{\min} = \pm \int_{\Omega} d^2 x \, \mathbf{m} \cdot (\partial_1 \mathbf{m} \times \partial_2 \mathbf{m}) \pm \oint_{\partial \Omega} dx_i \, \mathbf{A}_i \cdot \mathbf{m} = \pm 4\pi Q \pm \oint_{\partial \Omega} dx_i \, \mathbf{A}_i \cdot \mathbf{m}.
\tag{31}
$$

The first term in Eq. (31) is a topological invariant ($Q$ is the skyrmion number). The second, boundary term enforces gauge invariance of the energy (31). To see that, rotate the local spin frame by an infinitesimal angle $\boldsymbol{\omega}(x)$. The first, topological term is not sensitive to smooth changes of $\mathbf{m}(x)$ in the bulk of the region $\Omega$; all of its variation comes from the boundary $\partial \Omega$:

$$
\delta \int_{\Omega} d^2 x \, \mathbf{m} \cdot (\partial_1 \mathbf{m} \times \partial_2 \mathbf{m}) = -\oint_{\partial \Omega} dx_i \, \partial_i \boldsymbol{\omega} \cdot \mathbf{m}.
\tag{32}
$$

This dependence of the energy on the local spin frame is clearly unphysical. It is canceled by an equal and opposite variation of the boundary term:

$$
\delta \oint_{\partial \Omega} dx_i \, \mathbf{A}_i \cdot \mathbf{m} = \oint_{\partial \Omega} dx_i \, \partial_i \boldsymbol{\omega} \cdot \mathbf{m}.
\tag{33}
$$

The net energy (31) is then gauge-invariant.

The boundary term in Eq. (31) can be neglected in the thermodynamic limit. If $\mathbf{A}_i \cdot \mathbf{m}$ is bounded and the system has a finite skyrmion density $\rho = \frac{1}{4\pi} \mathbf{m} \cdot (\partial_1 \mathbf{m} \times \partial_2 \mathbf{m})$ then in the large-system limit the boundary term, scaling as the perimeter $|\partial \Omega|$ at most, will be much smaller than the bulk term, scaling as the area $|\Omega|$. The first condition always holds in our model, where $|\mathbf{m}| = 1$ and $\mathbf{A}_i = \mathbf{d}_i = \text{const}$ in a global frame. The second condition is true for skyrmion crystals.

In what follows, we take the upper sign in the Bogomolny equation (29). Its translation into the complex coordinates and fields (28) is given in Table 2. For the symmetry class $D_n$, the Bogomolny solutions have the form [12]

$$\bar{\chi} = w(z) + i\kappa\bar{z}/2. \tag{34}$$

The energy of these states is

$$E_{\min} = 4\pi Q + \oint_{\partial\Omega} dx_i\, \mathbf{A}_i \cdot \mathbf{m}. \tag{35}$$

### 3.2.1 False vacuum

Bogomolny solutions (34) include just one uniform state, $\bar{\chi} = \infty$, or $\mathbf{m}(x) = +\mathbf{e}^{(3)}$. It has $Q = 0$ and $E = 0$, so it is appropriate to call it the vacuum.

### 3.2.2 High-energy skyrmion crystal

Bogomolny solutions (34) also include skyrmion crystals. The function $w(z) = C/z$ yields a $Q = +1$ skyrmion with $\mathbf{m} = +\mathbf{e}^{(3)}$ at the origin. A skyrmion lattice is given by a meromorphic function $w(z)$ with periodically arranged simple poles. The Weierstrass $\zeta$ function [33] fits the bill. For a square lattice with the spatial period $a$, the $\zeta$ function has periods $2\omega_1 = a$ and $2\omega_2 = ia$. For a hexagonal lattice, $2\omega_1 = a$ and $2\omega_2 = ae^{2\pi i/3}$. The $\zeta$ function is not strictly periodic but quasiperiodic: $\zeta(z + 2\omega_n) = \zeta(z) + 2\eta_n$, $n = 1, 2$. (See Appendix B.) To make a periodic function, it suffices to add terms $z$ and $\bar{z}$ in the right proportions. The function

$$\bar{\chi} = \frac{i\kappa}{2}\left(\bar{z} - \frac{S}{\pi}\zeta(z)\right), \tag{36}$$

yields square and hexagonal skyrmion crystals with positive energy $E = 4\pi$ per unit cell (area $S \propto a^2$). See Appendix B for details. Earlier uses of the Weierstrass elliptic functions to describe soliton lattices can be found in Refs. [34, 35].

### 3.2.3 One antiskyrmion

To make the energy (31) negative, we may try states with negative skyrmion charge $Q$. Setting $w(z) = 0$ in Eq. (34), we obtain a Bogomolny state

$$\bar{\chi} = \frac{i\kappa\bar{z}}{2}, \quad \text{or} \quad \bar{\psi} = -\frac{2i}{\kappa z}, \tag{37}$$

with $Q = -1$. If the boundary term in Eq. (31) could be neglected, we would expect to find the energy $E = -4\pi$. However, the boundary term is not negligible. It adds $8\pi$ to the energy so that overall the energy $E = 4\pi$ is positive.

All is not lost, however. As we remarked earlier, the boundary term in Eq. (31) becomes negligible in the thermodynamic limit with a finite skyrmion density. If we could construct an antiskyrmion crystal ($Q = -1$ per unit cell) it would have the negative energy of $-4\pi$ per unit cell.

Alas, no such Bogomolny states exist. The lowest possible skyrmion number for a Bogomolny state (34), $Q = -1$, is achieved with $w(z) = 0$ (or any constant). If $w(z)$ is a polynomial of degree $N > 1$ then $Q = N > 1$.[5] We simply cannot construct a Bogomolny state with a skyrmion charge $Q < -1$. We have to look beyond Bogomolny states.

---

[5]For $N = 1$ it could be $-1$ or $+1$, depending on the relative amplitude of the $z$ and $\bar{z}$ terms.

### 3.2.4 Two antiskyrmions

We begin with a trial state

$$\bar{\psi} = -\frac{2i}{\kappa}\left(\frac{1}{z-a} + \frac{1}{z+a}\right). \tag{38}$$

It describes two antiskyrmions ($Q = -2$) distance $2a$ apart. In the limit of large separation, $2a \gg 1/\kappa$, the magnetization field near $z = a$ and $-a$ looks like a Bogomolny state (37), so we may expect that our trial state will describe two weakly interacting antiskyrmions.

Indeed, for $a \to \infty$ the energy of the trial state (38) is

$$E \sim 8\pi + \frac{512\pi}{(\kappa a)^2}\ln(C\kappa a), \tag{39}$$

with $C$ a numerical constant (see Appendix C for details). The leading term in Eq. (39) is $+8\pi$, rather than $-8\pi$, which may again be due to the boundary contribution in Eq. (35).[6] The second term represents a long-range repulsive interaction between antiskyrmions.

### 3.2.5 Well-separated antiskyrmions

It is now straightforward to construct a state with $N$ antiskyrmions:

$$\bar{\psi} = -\frac{2i}{\kappa}\sum_{n=1}^{N}\frac{1}{z-z_n}. \tag{40}$$

If the antiskyrmions are separated by distances much greater than $1/\kappa$ we expect their interactions to be weak and the energy to be $E \approx -4\pi N$ in the thermodynamic limit.

### 3.2.6 Antiskyrmion crystal

Our next step takes us go from a function with $N$ single poles (40) to one with an infinite periodic array of poles that would represent a crystal of antiskyrmions. The Weierstrass $\zeta$ function has a periodic lattice of poles [33]. In analogy with Eq. (36) for a skyrmion crystal, we write down an Ansatz for a (square or hexagonal) crystal of antiskyrmions:

$$\bar{\psi} = \frac{2i}{\kappa}\left(\frac{\pi}{S}\bar{z} - \zeta(z)\right). \tag{41}$$

The trial parameter here is the lattice constant $a$. In the limit $a \to \infty$, we recover the Bogomolny state with one antiskyrmion (37).

At large antiskyrmion separations $a \gg 2/\kappa$, the energy per unit cell asymptotically approaches $-4\pi$. By analogy with the $Q = -2$ case (39), we expect the leading correction to be $(\kappa a)^{-2}\ln(\kappa a)$. In the thermodynamic limit, the appropriate intensive variables are the (macroscopic) skyrmion and energy densities, $\rho = Q/S$ and $\mathcal{U} = E/S$:

$$\mathcal{U}(\rho) \sim 4\pi\rho - \frac{k\rho^2}{\kappa^2}\ln|C\kappa^2\rho|, \quad \rho \to -0. \tag{42}$$

Fig. 2 shows that the asymptotic form (42) describes the energy density of Ansatz (41) quite well. The curve $\mathcal{U}(\rho)$ has a minimum at the skyrmion density $\rho_0 = -0.0172\kappa^2$, where the negative energy of individual antiskyrmions is balanced by their repulsion. This corresponds to a hexagonal lattice of antiskyrmions with a lattice constant $a_0 = 8.19\kappa^{-1}$.

---

[6]Strictly speaking, Eq. (35) applies to Bogomolny states only.

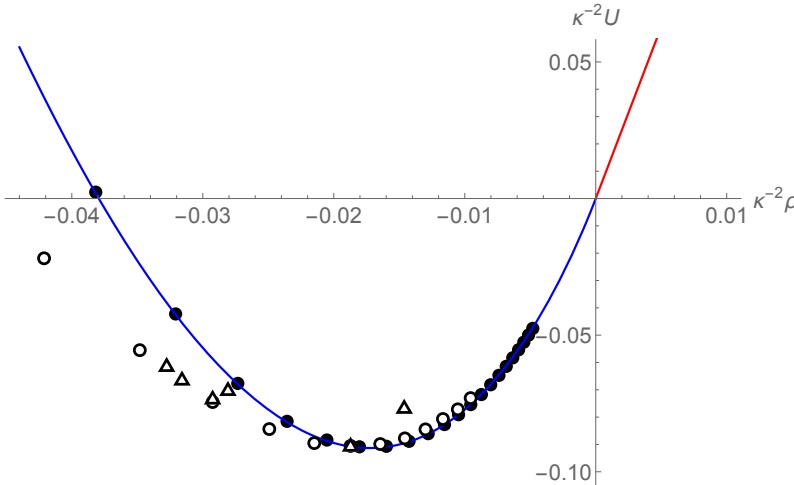

Figure 2: Energy density $\mathcal{U}$ of the skyrmion and antiskyrmion crystals vs. skyrmion density $\rho$. Lines are $4\pi\rho$ for the skyrmion crystal ($\rho > 0$, red) and Eq. (42) with $k = 115$ and $C = 1.47$ for the antiskyrmion crystal ($\rho < 0$, blue). Filled circles: trial state of Eq. (41) with a hexagonal antiskyrmion crystal. Open symbols: Monte Carlo simulations beginning with the following starting points. Open circles: trial state of Eq. (41). Open triangles: a random $T = \infty$ state.

To corroborate these theoretical results, we ran Monte Carlo simulations for a lattice version of the gauged Heisenberg model [28],

$$U = -J \sum_{\langle ij \rangle} \mathbf{S}_i \cdot R_{ij} \mathbf{S}_j - \gamma \sum_i \mathbf{h} \cdot \mathbf{S}_i. \tag{43}$$

It reproduces the continuum theory (23) with an appropriate choice of the exchange constant $J$ and the "gyromagnetic ratio" $\gamma$. For a hexagonal lattice of spins, $J = 1/\sqrt{3}$ and $\gamma = \sqrt{3}/2$. $R_{ij}$ is an SO(3) matrix with the angle of rotation $\kappa$ and the axis parallel to the link $\langle ij \rangle$ for the $D_n$ symmetry class. We worked on lattices with up to $30 \times 30$ sites and periodic boundary conditions.

The magic field in the lattice model is

$$\mathbf{h} = 4\mathbf{e}^{(3)} \sin^2(\kappa/2) \sim \kappa^2 \mathbf{e}^{(3)} \tag{44}$$

as $\kappa \to 0$. For $\kappa = \pi/6$, used in our simulations, the lattice and continuum magic fields differ by about 2%. This gives a rough estimate for the expected discrepancy between the lattice and continuum models. Monte Carlo simulations were run at a low temperature $T = 0.01$. For direct comparison with theory for the ground state, we subtracted the thermal energy of spin waves equal to $T$ per spin.

To check the local stability and accuracy of the antiskyrmion crystal Ansatz (41), we used this trial state as a starting point in the simulations. Topological stability of the skyrmion number enabled us to work at fixed skyrmion densities. The Ansatz turned out to be quite accurate when antiskyrmions are far apart, $|\rho| \ll \kappa^2$. The energy density of the final state (open circles in Fig. 2) agrees well with the theory predictions (filled circles) in the range $\rho = -0.02\kappa^2 \ldots - 0.01\kappa^2$, which includes the optimal density $\rho_0 \approx -0.0172\kappa^2$.

With the local stability of the antiskyrmion crystal confirmed, we have also searched for alternative ground states by starting simulations with a random high-energy ($T = \infty$) state and quenching it to $T = 0.01$. The final states (open triangles in Fig. 2) usually turned out to



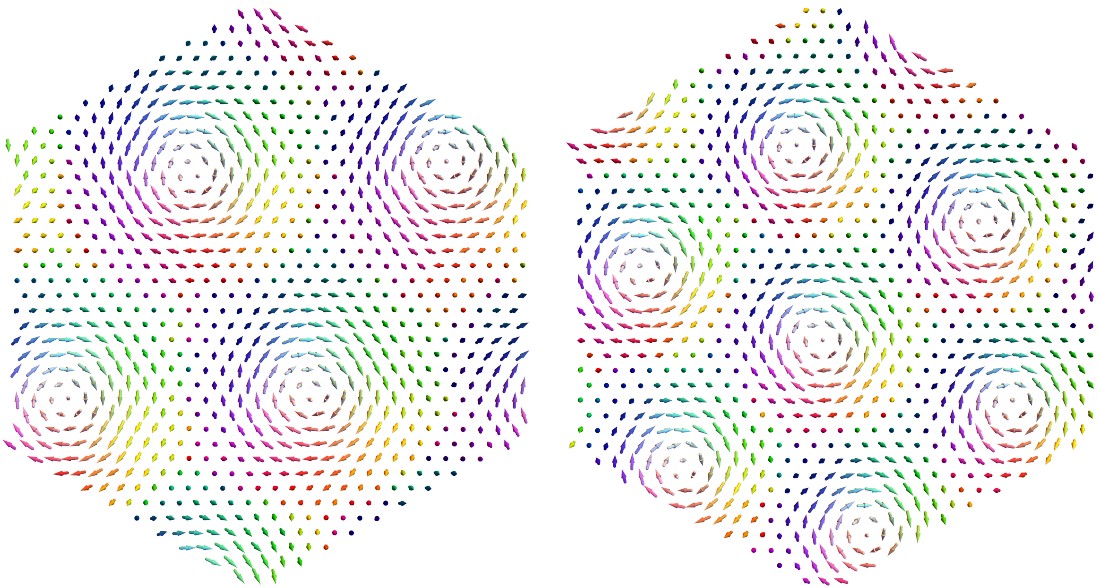

Figure 3: Spontaneous formation of hexagonal antiskyrmion crystals in the chiral lattice model (43) at $\mathbf{h} = +\kappa^2\mathbf{e}^{(3)}$ and $T = 0.01$. Monte Carlo simulations on a lattice with $30 \times 30$ sites and periodic boundary conditions. $\mathbf{m} = -\mathbf{e}^{(3)}$ at the centers of antiskyrmions and $+\mathbf{e}^{(3)}$ midway between them.

be the same hexagonal antiskyrmion crystals as those obtained by starting from Ansatz (41). Fig. 3 shows two such final states. In other cases, the magnet was trapped in metastable states of higher energy.

It is thus reasonable to conclude that the ground state of the gauged Heisenberg ferromagnet (23) in the magic field $\mathbf{h} = \kappa^2\mathbf{e}^{(3)}$ is the hexagonal antiskyrmion crystal with the lattice constant $a_0 \approx 8.19\kappa^{-1}$.

# 4 Conclusion

We have discussed a geometrized version of the Heisenberg model, whose energy remains invariant under local spin-frame rotations. The possible existence of a twist between spin frames at adjacent spatial points necessitates the introduction of the spin connection, or the SO(3) gauge field. This is entirely analogous to the geometrization of gravity in Einstein's relativity. It is remarkable that the Dzyaloshinskii-Moriya (DM) interaction, responsible for chiral magnetism, becomes a geometric effect in this theory, with the DM vectors playing the role of the SO(3) gauge field. As such, the DM interaction itself is not a physical quantity because it is spin-frame-dependent and can even be gauged away for a chosen spatial direction. The closest related physical observable is the curvature of the gauge field, which is quadratic in the DM couplings.

We relied on this geometric perspective to find the ground state of a chiral ferromagnet in a magic magnetic field equal to the spin curvature. At that special point, one can use the Bogomolny approach to obtain exact configurations $\mathbf{m}(x)$ locally minimizing the energy. It is unfortunate that the absolute ground state lies in a topological sector with no Bogomolny states. However, we were able to construct trial states in the form of an antiskyrmion crystal that very likely are a good approximation for the absolute ground state.

The antiskyrmion crystal remains locally stable in a range of applied fields. Its energy

remains below the vacuum level even as the applied field is increased slightly or reduced to zero. At zero field, the antiskyrmion crystal coexists with a skyrmion crystal, in agreement with the time-reversal symmetry.

Beyond this specific application, the geometrized theory of chiral magnetism offers an interesting new perspective. For example, various physical observables—e.g., magnetization $\mathbf{m}(x)$, spin current $\mathbf{j}_i(x)$, and spin curvature $\mathbf{F}_{ij}(x)$—transform like vectors under local rotations of the spin frame, with the law of transformation given by Eq. (3). In contrast, the DM vectors are an SO(3) gauge field $\mathbf{A}_i(x) = \mathbf{d}_i$ that transforms differently, according to Eq. (6). This difference in symmetry properties suggests that DM vectors $\mathbf{d}_i$ cannot be related by a linear proportionality to physical spin vectors such as spin current $\mathbf{j}_i(x)$. Precisely such a linear relationship between spin current and DM interactions has been suggested recently in theoretical proposals [36–38] and experimental studies [39, 40]. A critical reexamination of the proposed relation may be in order.

# Acknowledgements

The authors thank Sayak Dasgupta, Se Kwon Kim, Volodymyr Kravchuk, Predrag Nikolic, Zohar Nussinov, Yuan Wan, and Shu Zhang for discussions. D.H. and O.T. have been supported by the US DOE Basic Energy Sciences, Materials Sciences and Engineering Award DE-SC0019331. This work has been done in part at the Aspen Center for Physics supported by the US NSF Grant PHY-1607611 and at the Kavli Institute for Theoretical Physics, supported by the US NSF Grant PHY-1748958. V.S. thanks the Max Planck Institute for Mathematics in the Sciences, Leipzig, for hospitality and acknowledges support by the Leverhulme grant RPG-2018-438.

# A   Examples of an SO(3) gauge field

## A.1   Gauge field from spatial twists of the local spin frame

The local frame at a spatial point $x$ is specified by $N$ mutually orthogonal unit vectors $\hat{\mathbf{e}}^{(\alpha)}(x)$, $\alpha = 1, 2, \ldots, N$. As in the main text, the boldface symbol $\hat{\mathbf{e}}^{(\alpha)}$ is a shorthand for $N$ components $(e_{\mathrm{I}}^{(\alpha)}, e_{\mathrm{II}}^{(\alpha)}, \ldots, e_{\mathrm{N}}^{(\alpha)})$. Unlike in the main text, these components are specified in a fixed global spin frame and are labeled by Roman numerals for distinction.

The unit vectors of the local frame satisfy the relations

$$\hat{\mathbf{e}}^{(\alpha)}(x) \cdot \hat{\mathbf{e}}^{(\beta)}(x) = \delta_{\alpha\beta}, \quad \hat{\mathbf{e}}^{(\alpha)}(x) \wedge \hat{\mathbf{e}}^{(\beta)}(x) \wedge \ldots \wedge \hat{\mathbf{e}}^{(\nu)}(x) = \epsilon_{\alpha\beta\ldots\nu}. \tag{45}$$

Here all unit vectors are taken at the same spatial point $x$; $\epsilon_{\alpha\beta\ldots\nu}$ is the antisymmetric Levi-Civita symbol in $N$ dimensions.

The orthogonality relations will generally not hold for unit vectors $\hat{\mathbf{e}}^{(\alpha)}(x)$ and $\hat{\mathbf{e}}^{(\beta)}(x')$ at different points if the local frame twists in space. The frame twists define a spin connection $a_{i\alpha\beta}$ for spatial direction $i$:

$$a_{i\alpha\beta} \equiv \hat{\mathbf{e}}^{(\alpha)} \cdot \partial_i \hat{\mathbf{e}}^{(\beta)} = -\hat{\mathbf{e}}^{(\beta)} \cdot \partial_i \hat{\mathbf{e}}^{(\alpha)} \equiv -a_{i\beta\alpha}. \tag{46}$$

For a given $i$, the coefficients $a_{i\alpha\beta}$ form matrix elements of an antisymmetric $N \times N$ matrix $a_i$. Such matrices are generators of the SO($N$) group.

The gauge curvature is an antisymmetric $N \times N$ matrix

$$f_{ij} = \partial_i a_j - \partial_j a_i + [a_i, a_j]. \tag{47}$$

It is straightforward to check that the gauge field (46) is trivial, $f_{ij} = 0$.

The above construction works for spin spaces with any number of dimensions $N$. For $N = 3$, an antisymmetric matrix has 3 independent components and therefore can be expressed in terms of a dual vector:

$$a_{i\alpha\beta} = \epsilon_{\alpha\beta\gamma}A_{i\gamma}, \quad f_{ij\alpha\beta} = \epsilon_{\alpha\beta\gamma}F_{ij\gamma}. \tag{48}$$

This introduces vector-valued SO(3) gauge field $\mathbf{A}_i$ and its curvature

$$\mathbf{F}_{ij} = \partial_i\mathbf{A}_j - \partial_j\mathbf{A}_i - \mathbf{A}_i \times \mathbf{A}_j. \tag{49}$$

### A.2 Gauge field describing nontrivial parallel transport

Consider now a different example, where the SO(3) gauge field describes a nontrivial parallel transport.

We will work with the local spin frame that has a fixed orientation relative to a global spin frame, so there is no gauge field associated with twists of the local frame (Sec. A.1).

Suppose the rules of parallel transport are as follows: a spin being transported along the spatial axis $x_i$ physically rotates at a fixed spatial rate $\mathbf{k}_i$:

$$\partial_i\mathbf{m} = \mathbf{k}_i \times \mathbf{m}. \tag{50}$$

These rules are equivalent to the condition $D_i\mathbf{m} = 0$ with the SO(3) gauge field $\mathbf{A}_i = \mathbf{k}_i$.

If the wavevectors $\mathbf{k}_i$ are noncollinear then the parallel transport is nontrivial as it has nonzero curvature

$$\mathbf{F}_{ij} = -\mathbf{k}_i \times \mathbf{k}_j \neq 0. \tag{51}$$

## B  Weierstrass zeta function

### B.1 Definition

The Weierstrass $\zeta$ function is a meromorphic function of the complex variable $z$ with single poles of unit residue arranged in a periodic manner in the complex plane [33]. Fundamental complex periods $2\omega_1$ and $2\omega_2$ define the locations of all the poles

$$\Omega_{mn} = 2m\omega_1 + 2n\omega_2, \tag{52}$$

where $m$ and $n$ are arbitrary integers.

Taking $2\omega_1 = a$ and $2\omega_2 = ia$ yields a square lattice of poles. A hexagonal lattice obtains for $2\omega_1 = a$ and $2\omega_2 = ae^{2i\pi/3}$. The (oriented) area of a unit cell can be written as

$$S = 2i(\omega_1\bar{\omega}_2 - \omega_2\bar{\omega}_1) = 4|\omega_1||\omega_2|\sin(\phi_2 - \phi_1), \quad \phi_n = \arg\omega_n. \tag{53}$$

It is customary to label the fundamental periods in such a way that the area $S$ is positive.

The Weierstrass $\zeta$ function is defined as

$$\zeta(z) = \frac{1}{z} + \sum_{m,n}{}' \left( \frac{1}{z - \Omega_{mn}} + \frac{1}{\Omega_{mn}} + \frac{z}{\Omega_{mn}^2} \right), \tag{54}$$

where the prime signifies the exclusion of the pair $m = n = 0$ from the sum. The function is odd, $\zeta(-z) = -\zeta(z)$. It behaves as as $\zeta(z) = 1/z + \mathcal{O}(z^3)$ near $z = 0$.

## B.2 Quasiperiodicity

Although the poles form a periodic lattice, the $\zeta$ function is not strictly periodic. Rather, it is quasiperiodic:

$$\zeta(z + 2\omega_n) = \zeta(z) + 2\eta_n, \quad n = 1, 2. \tag{55}$$

The quasiperiodicity parameters $\eta_n = \zeta(\omega_n)$ satisfy a fundamental relation

$$\eta_1 \omega_2 - \eta_2 \omega_1 = i\pi/2. \tag{56}$$

The quasiperiodicity of the $\zeta$ function can be understood by using an analogy with electrostatics in 2 spatial dimensions. An electrostatic field $\mathbf{E}(x, y)$ maps to a meromorphic function $\zeta(z)$ as follows [41]:

$$z = x + iy, \quad \zeta = E_x - iE_y. \tag{57}$$

A pole maps to an electric charge, the pole's residue gives the electric charge. The Weierstrass $\zeta$ function with periodic poles (52) maps to a crystal of unit electric charges. Although the charge distribution is spatially periodic, the resulting electric field is not. Because of the long-range nature of the Coulomb force, the distribution of the electric field near the center of the crystal is different from the distribution near the edge. A related problem is the calculation of the Madelung constant in ionic crystals [42].

## B.3 Constructing a periodic function

To obtain a truly periodic function, we may use a linear combination of $\zeta(z)$ with $z$ and $\bar{z}$,

$$f(z, \bar{z}) = \zeta(z) + Az + B\bar{z}. \tag{58}$$

From

$$f(z + 2\omega_n, \bar{z} + 2\bar{\omega}_n) = f(z, \bar{z}) + 2\eta_n + 2\omega_n A + 2\bar{\omega}_n B, \quad n = 1, 2, \tag{59}$$

we find that the strict periodicity is achieved provided that

$$\eta_n + \omega_n A + \bar{\omega}_n B = 0, \quad n = 1, 2. \tag{60}$$

These linear equations have a unique solution,

$$A = -\frac{\eta_1 \bar{\omega}_2 - \eta_2 \bar{\omega}_1}{\omega_1 \bar{\omega}_2 - \omega_2 \bar{\omega}_1} = \frac{2(\eta_1 \bar{\omega}_2 - \eta_2 \bar{\omega}_1)}{iS}, \quad B = \frac{\eta_1 \omega_2 - \eta_2 \omega_1}{\omega_1 \bar{\omega}_2 - \omega_2 \bar{\omega}_1} = -\frac{\pi}{S}, \tag{61}$$

where we have used identities (53) and (56).

For a square lattices of poles, the quasiperiodicity parameters are [43]

$$2\eta_1 = -\frac{i\pi}{2\bar{\omega}_2}, \quad 2\eta_2 = -\frac{i\pi}{2\bar{\omega}_1}. \tag{62}$$

These yield $A = 0$, so that the periodic function is a linear combination of $\zeta(z)$ and $\bar{z}$:

$$f(z, \bar{z}) = \zeta(z) - \frac{\pi}{S}\bar{z}, \tag{63}$$

where $S = a^2$ is the area of the unit cell.

Eq. (63) also applies to the hexagonal lattice with $S = a^2 \sqrt{3}/2$. For a generic lattice, the coefficient $A$ does not vanish, so the periodic function $f(z, \bar{z})$ has an admixture of $z$.

Function (63) is shown in color plots of Fig. 4 for the square and hexagonal lattices.

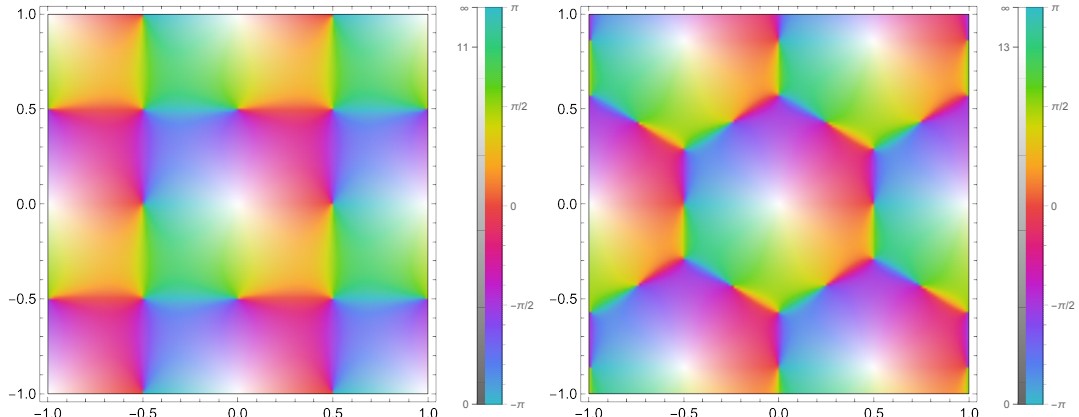

Figure 4: Color plots of the complex function $f$ (63) for the square and hexagonal lattices with $a = 1$. Brightness encodes the magnitude: $|f| = 0$ is black, $\infty$ is white. Hue indicates the phase: $\arg f = 0$ is red, $2\pi/3$ is green, $4\pi/3$ is blue.

## C   Interaction energy of two antiskyrmions

In this section we evaluate the interaction energy of two antiskyrmions as a function of their distance. To that end, we use the trial state (38). The energy density at the magic field is

$$\mathcal{U} = \frac{1}{2}D_i\mathbf{m} \cdot D_i\mathbf{m} - \kappa^2 \mathbf{e}^{(3)} \cdot \mathbf{m} = \frac{64}{\kappa^2}\frac{a^4 + 5r^4 - 2a^2r^2\cos 2\phi}{(a^4 + r^4 - 2a^2r^2\cos 2\phi + 16\kappa^{-2}r^2)^2}. \tag{64}$$

Averaging the energy density over the azimuthal angle $\phi$ yields

$$\bar{\mathcal{U}}(r) = \frac{64}{\kappa^2}\frac{a^8 + 2a^4r^4 + 5r^8 + 16\kappa^{-2}r^2(a^4 + 5r^4)}{[(a^4 - r^4)^2 + 32\kappa^{-2}r^2(a^4 + r^4) + 256\kappa^{-4}r^4]^{3/2}}. \tag{65}$$

The energy is

$$U(a) = 2\pi \int_0^\infty r\,dr\,\bar{\mathcal{U}}(r). \tag{66}$$

We obtain the asymptotic behavior of the energy in the limit of large separation between the antiskyrmions, $2a \gg \kappa^{-1}$. The integrand $r\bar{U}(r)$ is peaked near $r = a$. As $a \to \infty$, both the height and width of the peak remain $\mathcal{O}(a^0)$. Treating $r\tilde{U}(r)$ as a function of $\xi \equiv \kappa(r - a)$ and expanding it in inverse powers of $a$, we obtain at the zeroth order its asymptotic shape,

$$\bar{\mathcal{U}}(r)r\,dr \sim \frac{8\,d\xi}{(\xi^2 + 4)^{3/2}}, \quad a \to \infty. \tag{67}$$

Integrating over the area yields

$$U(a) \sim 2\pi \int_{-\kappa a}^\infty \frac{8\,d\xi}{(\xi^2 + 4)^{3/2}} \sim 2\pi \int_{-\infty}^\infty \frac{8\,d\xi}{(\xi^2 + 4)^{3/2}} = 8\pi\,, \tag{68}$$

as $a \to \infty$. Here we expanded the integration range to the entire $\xi$ axis as the integral converges. Again, we find that the energy of two distant antiskyrmions tends to $+8\pi$, rather than $-8\pi$. No worries: two antiskyrmions in an infinite plane is not the thermodynamic limit.

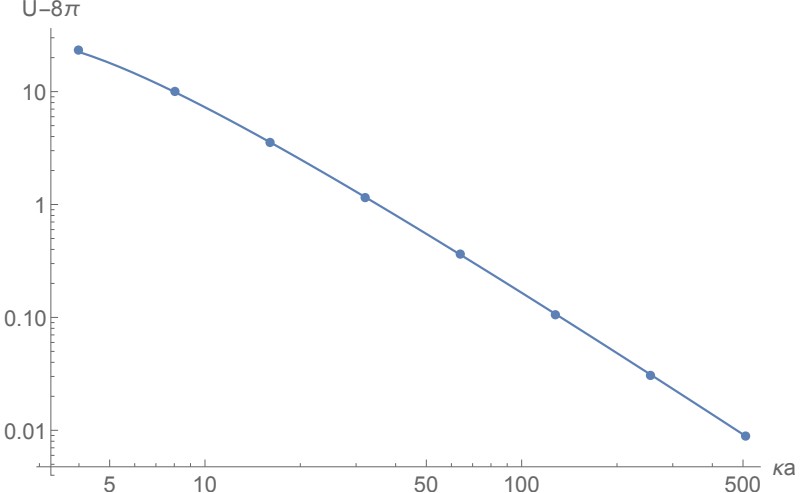

Figure 5: Interaction energy of two antiskyrmions distance $2a$ apart. The points are obtained by numerical integration of the potential energy density (65). The line is the asymptotic form (70) with $C = 0.61$.

Expanding to higher orders in $a^{-1}$ yields

$$\bar{U}(r)r\,dr = \frac{8\,d\xi}{(\xi^2+4)^{3/2}} + \frac{1}{\kappa a}\frac{4\xi(5\xi^2+8)d\xi}{(\xi^2+4)^{5/2}} + \frac{1}{(\kappa a)^2}\frac{4(8\xi^6+49\xi^4+176\xi^2+192)d\xi}{(\xi^2+4)^{7/2}} + \cdots \tag{69}$$

The term $\mathcal{O}(a^{-1})$ is odd in $\xi$ and vanishes upon integration (over the entire $\xi$ axis). Therefore the $\mathcal{O}(a^{-1})$ correction to the energy vanishes.

The computation of the $\mathcal{O}(a^{-2})$ correction runs into a problem. The last term behaves asymptotically as $32\,d\xi/|\xi|$ as $\xi \to \infty$, so the integral over $\xi$ diverges if we set the limits to $\pm\infty$. However, we already know that the lower limit is finite, $-\kappa a$, so it is natural to cut the integration at $\kappa a$ at the upper limit as well. With this assumption, the asymptotic behavior of the energy is

$$U(a) \sim 8\pi + \frac{128\pi}{(\kappa a)^2}\ln(C\kappa a), \tag{70}$$

where $C$ is a numerical constant. The second term represents the interaction energy of two antiskyrmions distance $2a$ apart. Fig. 5 shows that the asymptotic form works well in the range of distances $\kappa a$ from 4 to 512.

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
