# Peer review of "Chiral magnetism: a geometric perspective"

_SciPost Physics, doi:SciPost Phys. 10, 078 (2021)_

## Round 4 · Referee Report · Anonymous (Referee 1) · 2021-3-5

Report

This paper introduces a geometric picture of the Dzyaloshinskii-Moriya (DM) interaction as a background gauge field, and uses this picture to construct exact skyrmion latttice states that are the ground state of the system for a particular field magnitude, which is corroborated by classical Monte Carlo calculations. The paper provides a new synergetic link between the ideas of parallel transport on curved manifolds and the theory of DM interactions, and is extremely clear and well written for a broad audience. I believe it should be published after the points below are addressed.

  • I have one point of confusion: is the spin current a physical spin vector or not? The conclusion on page 16 states that it is, but unless I’m mistaken equation (18) does not transform like a spin vector (given that \partial_i m does not). There would then be no inconsistency with references 34-38 and the whole final paragraph is no longer relevant.

  • It is not obvious from this paper that ref. [34] introduced the expression of the DM interaction in terms of the covariant derivative, even in the historical discussion and I think that it should be. Section 2.4 should be modified to make it clear what is novel and what is not.

  • At the beginning of Section 3, it is stated that the “ground state of the model is a hexagonal skyrmion crystal”, but this statement appears to be overly broad and it is only the ground state in/possibly near the magic magnetic field. This should be clarified here.

  • validity: top
  • significance: high
  • originality: high
  • clarity: top
  • formatting: perfect
  • grammar: excellent

Author:  Oleg Tchernyshyov  on 2021-03-16  [id 1314]

(in reply to Report 1 on 2021-03-05)

We thank the referee for careful reading of the manuscript and for raising thoughtful questions. They are answered below.

  • Eq. (18) is the commonly accepted definition of spin current. As the referee correctly points out, it does not transform like a spin vector. A revised definition of the spin current, where the ordinary gradient is replaced the covariant derivative, is given below Eq. (20). The redefined spin current does transform like a spin vector. To make this point clear, we put the gauge-invariant definition of the spin current as displayed Eq. (21) and added "gauge-invariant" to its characterization immediately thereafter.

  • Eq. (3) in Ref. 34 looks superficially like a covariant derivative, but on closer inspection it isn't one. The role of the gauge potential there is taken by an external spin current. A spin current is a physical spin vector (as has been mentioned in the previous query) and the gauge potential isn't, so the former cannot simply replace the latter in a covariant derivative. For this reason, Eq. (3) does not represent a covariant derivative. In fact, we point out this paradox in the Conclusion.
    Priority in treating the DM interaction as an SO(3) gauge field isn't so easy to settle. Hints have appeared in earlier papers. Gaidedei, Sheka et al. [22,23] used a covariant formulation of the Heisenberg model on a curved surface to reveal "curvature induced effective Dzyaloshinskii-like interaction." Much earlier, Shekhtman et al. [16] reversed-engineered the DM interaction in a lattice model to show that it can be written as a gauged Heisenberg model. None of these works, including Ref. 34, articulated clearly that the DM interaction can be thought of as a geometrical effect of spin parallel transport. As far as we know, Schroers [11,13] was the first to make this connection clear. We mention all of the above works, and more, in our historical review, but feel that Schroers deserves the most credit for the insight, as do Fröhlich and Studer [17] for their realization of the same for the case of itinerant electrons and an SU(2) gauge field.

  • We have clarified this overly broad statement about the nature of the ground state by adding the qualifier "at least for a magic value of an applied magnetic field and possibly beyond."

---

## Round 4 · Referee Report · Anonymous (Referee 2) · 2021-3-7

Strengths

The manuscript under review discusses the geometric perspective on chiral ferromagnetism whereby Dzyaloshinskii-Moriya interaction can be viewed as arising from the background non-abelian SO(3) field produced by DM vectors. The perspective is not entirely new but is explained particularly well in the present submission. I believe this makes the manuscript one of the nice “go-to” resources for novice readers on the topic, such as beginning graduate students.

Weaknesses

None

Report

The key finding of the manuscript is the proof, involving analytical argument and numerical simulations, that for the special value of the Zeeman magnetic field the ground state of the gauged Heisenberg ferromagnet is the antiskyrmion crystal. That special value of the field is given by the gauge curvature, eq.23. In this case, the stationary configuration of the magnetization m(x) satisfies the first-order differential equation (Bogomolny).

An agreement between this analytical prediction and Monte Carlo simulations of the lattice gauged theory is quite good. This is an interesting and new result.

It is interesting to ask what is the range of stability of the found antiskyrmion crystal with respect to the external magnetic field. The “soluble” point corresponds to one specific value of the field, eq.43. Could the authors comment in what range of fields around this value does the crystal remain a good approximation to the ground state?
  • validity: high
  • significance: high
  • originality: high
  • clarity: top
  • formatting: excellent
  • grammar: excellent

Author:  Oleg Tchernyshyov  on 2021-03-16  [id 1315]

(in reply to Report 2 on 2021-03-07)

Indeed, we have investigated stability of the antiskyrmion crystal state beyond the magic value of the magnetic field. The antiskyrmion crystal remains locally stable in a range of applied fields. Its energy remains below the vacuum level even as the applied field is increased slightly or reduced to zero. At zero field, the antiskyrmion crystal coexists with a skyrmion crystal, in agreement with the time-reversal symmetry. This clarification has been added to the paper's Conclusion.

---

## Round 4 · Referee Report · Anonymous (Referee 3) · 2021-3-10

Strengths

  1. Very clear narrative and motivation

  2. Addresses a specific problem of finding a (variational) ground state for a skyrmion crystal

  3. Reminds the audience of the geometrical approach used for non-collinear interactions

Weaknesses

none identified

Report

The manuscript addresses the problem of finding a variational ground state for a skyrmion crystal in 2D Heisenberg systems with Dzyaloshinskii-Moriya interactions. The approach uses an interesting approach that includes the DM interaction as a gauge potential and sees the resulting chiral order as the system's response to the local geometry and the associated parallel spin transport. This geometrical approach is elegant and intuitive and allows for the natural generalization of the Bogomolny equations in the case of a gauge Heisenberg interaction.

The paper is very well written, providing appropriate “historical” background before launching into the specific solution of a skyrmion crystal. That section is also rather pedagogical and clear. [There is a typo in the word “Seting” just before eq. 36, as well as a missing definition for eta_n right before eq. 35 (or link to the appendix).]

This paper provides the community with a welcome readable reminder of an elegant and attractive approach to dealing with non-collinear interactions. The practical insights this approach provides should guide the search for variational states for non-trivial geometries or other interactions. Moreover, the overall physical framework is worth keeping in mind when analyzing chiral systems.

Requested changes

Minor typos as indicated

---

## Round 5 · Author Response

Dear Dr. Sandler,

We are pleased to learn that the referees gave our manuscript favorable reviews. Below we address their request for minor changes. We have made a few minor edits as a result of these queries and added two references of prior use of the Weierstrass elliptic functions. We hope that the revision will make the manuscript suitable for publication in SciPost Physics.

With best regards,

Daniel Hill Valeriy Slastikov Oleg Tchernyshyov

Referee 1

  • I have one point of confusion: is the spin current a physical spin vector or not? The conclusion on page 16 states that it is, but unless I’m mistaken equation (18) does not transform like a spin vector (given that \partial_i m does not). There would then be no inconsistency with references 34-38 and the whole final paragraph is no longer relevant.

Eq. (18) is the commonly accepted definition of spin current. As the referee correctly points out, it does not transform like a spin vector. A revised definition of the spin current, where the ordinary gradient is replaced the covariant derivative, is given below Eq. (20). The redefined spin current does transform like a spin vector. To make this point clear, we put the gauge-invariant definition of the spin current as displayed Eq. (21) and added "gauge-invariant" to its characterization immediately thereafter.

  • It is not obvious from this paper that ref. [34] introduced the expression of the DM interaction in terms of the covariant derivative, even in the historical discussion and I think that it should be. Section 2.4 should be modified to make it clear what is novel and what is not.

Eq. (3) in Ref. 34 looks superficially like a covariant derivative, but on closer inspection it isn't one. The role of the gauge potential there is taken by an external spin current. A spin current is a physical spin vector (as has been mentioned in the previous query) and the gauge potential isn't, so the former cannot simply replace the latter in a covariant derivative. For this reason, Eq. (3) does not represent a covariant derivative. In fact, we point out this paradox in the Conclusion.

Priority in treating the DM interaction as an SO(3) gauge field isn't so easy to settle. Hints have appeared in earlier papers. Gaidedei, Sheka et al. [22,23] used a covariant formulation of the Heisenberg model on a curved surface to reveal "curvature induced effective Dzyaloshinskii-like interaction." Much earlier, Shekhtman et al. [16] reversed-engineered the DM interaction in a lattice model to show that it can be written as a gauged Heisenberg model. None of these works, including Ref. 34, articulated clearly that the DM interaction can be thought of as a geometrical effect of spin parallel transport. As far as we know, Schroers [11,13] was the first to make this connection clear. We mention all of the above works, and more, in our historical review, but feel that Schroers deserves the most credit for the insight, as do Fröhlich and Studer [17] for their realization of the same for the case of itinerant electrons and an SU(2) gauge field.

  • At the beginning of Section 3, it is stated that the “ground state of the model is a hexagonal skyrmion crystal”, but this statement appears to be overly broad and it is only the ground state in/possibly near the magic magnetic field. This should be clarified here.

We have clarified this statement by adding the qualifier "at least for a magic value of an applied magnetic field and possibly beyond."

Referee 2

  • It is interesting to ask what is the range of stability of the found antiskyrmion crystal with respect to the external magnetic field. The “soluble” point corresponds to one specific value of the field, eq.43. Could the authors comment in what range of fields around this value does the crystal remain a good approximation to the ground state?

We added the following clarification to the Conclusion. "The antiskyrmion crystal remains locally stable in a range of applied fields. Its energy remains below the vacuum level even as the applied field is increased slightly or reduced to zero. At zero field, the antiskyrmion crystal coexists with a skyrmion crystal, in agreement with the time-reversal symmetry."

Referee 3

  • There is a typo in the word “Seting” just before eq. 36, as well as a missing definition for eta_n right before eq. 35 (or link to the appendix).

We fixed the typo and added a pointer to Appendix B.

---

## Round 5 · List of Changes

Put the gauge-invariant definition of the spin current as displayed Eq. (21) and added "gauge-invariant" to its characterization immediately thereafter.

Clarified the statement that the “ground state of the model is a hexagonal skyrmion crystal” by adding the qualifier "at least for a magic value of an applied magnetic field and possibly beyond."

Added the following clarification to the Conclusion. "The antiskyrmion crystal remains locally stable in a range of applied fields. Its energy remains below the vacuum level even as the applied field is increased slightly or reduced to zero. At zero field, the antiskyrmion crystal coexists with a skyrmion crystal, in agreement with the time-reversal symmetry."

Fixed the typo “Seting” just before eq. 36.

Added a pointer to Appendix B for the definition of eta_n right before eq. 35.

Added ref [35]: D. Capic, et al, Phys. Rev. Research 1, 033011 (2019)

---

## Editorial Decision

published